# Exponential Lower Bounds for Fictitious Play in Potential Games

**Ioannis Panageas**[*]
University of California, Irvine
ipanagea@ics.uci.edu

**Nikolas Patris**[*]
University of California, Irvine
npatris@uci.edu

**Stratis Skoulakis**[*]
LIONS, EPFL
efstratios.skoulakis@epfl.ch

**Volkan Cevher**
LIONS, EPFL
volkan.cevher@epfl.ch

## Abstract

Fictitious Play (FP) is a simple and natural dynamic for repeated play with many applications in game theory and multi-agent reinforcement learning. It was introduced by Brown [4, 5] and its convergence properties for two-player zero-sum games was established later by Robinson [22]. Potential games [19] is another class of games which exhibit the FP property [18], i.e., FP dynamics converges to a Nash equilibrium if all agents follows it. Nevertheless, except for two-player zero-sum games and for specific instances of payoff matrices [1] or for adversarial tie-breaking rules [9], the *convergence rate* of FP is unknown. In this work, we focus on the rate of convergence of FP when applied to potential games and more specifically identical payoff games. We prove that FP can take exponential time (in the number of strategies) to reach a Nash equilibrium, even if the game is restricted to two agents and for arbitrary tie-breaking rules. To prove this, we recursively construct a two-player coordination game with a unique Nash equilibrium. Moreover, every approximate Nash equilibrium in the constructed game must be close to the pure Nash equilibrium in $\ell_1$-distance.

## 1 Introduction

In 1949 Brown [4, 5] proposed an uncoupled dynamics called *fictitious play* so as to capture the behavior of selfish agents engaged in a repeatedly-played game. Fictitious play assumes at round $t = 1$, each agent selects an arbitrary action. At each round $t \geq 2$, each player plays a best response pure action to the opponents' *empirical strategy*; empirical strategy is defined to be the average of the past chosen strategies.

Due to its simplicity and natural behavioral assumptions, fictitious play is one of the most seminal and well-studied game dynamics [7]. Despite the fact that fictitious play does not converge to a Nash equilibrium (NE) in general normal-form games, there are several important classes of games at which the *empirical strategies* always converge to a NE. In her seminal work, Robinson [22] showed that in the case of *two-player zero-sum* games, the empirical strategy profiles converge to a min-max equilibrium of the game; Robinson's proof use a smart inductive argument on the number of strategies of the game. Later, Monderer and Shapley [18] established that in the case of $N$-player potential games, the empirical strategies also converge to a NE. Summing up, the following theorem is true:

---

[*]Equal contribution.

37th Conference on Neural Information Processing Systems (NeurIPS 2023).

**Informal Theorem** [22],[18] *In two-player zero-sum and $N$-player potential games, the empirical strategy profiles of fictitious play converge to a NE for any initialization and tie-breaking rule*[*].

The above convergence results are asymptotic in the sense that they do not provide guarantees on the number of rounds needed by fictitious play to reach an approximate NE. Karlin [13] conjectured that in the case of two-player zero-sum games, fictitious play requires $O(1/\epsilon^2)$ rounds to reach an $\epsilon$-approximate NE. Daskalakis et. al. [9] disproved the strong version of Karlin's conjecture by providing an adversarial tie-breaking rule for which fictitious play requires exponential number of rounds (with respect to the number of strategies) in order to converge to an $\epsilon$-approximate NE. On the other hand, for the case of potential games and despite the foundational work of Monderer et al. [18], no lower bound results have been established. Therefore, the central focus of our work is to address the following question."

*Q: Does fictitious play in potential games admit convergence to an approximate NE with rates that depend polynomially on the number of actions and the desired accuracy?*

**Our contributions**   Our work provides a negative answer on the above question. Specifically, we present a *two-player potential* game for which fictitious play requires super-exponential time with respect to the number of actions to reach an approximate NE.

**Theorem 1.1** (Main result, formally stated Theorem 3.1). *There exists a two-player potential game (more specifically both agents have identical payoffs) in which both agents admit $n$ actions and for which fictitious play requires $\Omega\left(4^n\left(\left(\frac{n}{2}-2\right)!\right)^4 + \frac{1}{n\sqrt{\epsilon}}\right)$ rounds in order to reach an $\epsilon$-approximate NE. Moreover, the result holds for any tie-breaking rule and uniformly random initialization.*

**Remark 1.2.** Daskalakis et al. [9] provide an exponential lower bound on the convergence of fictitious play assuming an adversarial tie-breaking rule, meaning that ties are broken in favor of slowing down the convergence rate. To this point, it is not known whether fictitious play with a consistent tie-breaking rule (e.g. lexicographic) converges in polynomial time or not, except for special cases, e.g., diagonal payoff matrices [1]. We would like to note that our lower bound construction holds for any tie-breaking rule. The latter indicates an interesting discrepancy between two-player zero-sum and two-player potential games.

**Related Work**   The work of Daskalakis et al. [9] provides a lower bound on the convergence rate of fictitious play for the case of two-player zero-sum games using an adversarial tie-breaking rule and is the most closely related to our. Another important work is [1], in which they demonstrate that when the tie-breaking rule is predetermined (e.g., using lexicographical tie-breaking), the convergence rate of fictitious play becomes polynomial in the number of strategies, specifically with a rate of $O\left(\frac{1}{\epsilon^2}\right)$, as Karlin's conjecture states. It is important to note that this result holds true only for diagonal matrices.

Other works include the examination of fictitious play convergence for near-potential games [6], and the investigation of the necessary conditions that games must meet for fictitious play to converge to a NE [8]. We note that neither of these works provide convergence rates for fictitious play. On the other hand, fast convergence rates of *continuous time* fictitious play for *regular*, as introduced by Harsanyi [10], potential games are established [25]. Additional works on continuous-time fictitious play can be found in [21], and references there in.

Fictitious play has found various application in Multi-agent Reinforcement Learning as well (see [20, 23, 2, 3] and [26] for a survey and references therein), extensive form games [11], and control theory [15, 14]. It possesses not only an intuitive game theoretic interpretation but also other attractive properties, such as its simplicity with no need for a step-size parameter. These traits have contributed to its status as a pivotal topic of study, leading to numerous works aiming to establish convergence in broader settings [16, 24, 17, 12].[*]

**Technical overview**   The technical overview of this paper provides a high-level roadmap of the key contributions. In Section 3, we outline the steps towards proving Theorem 3.1 by recursively constructing a payoff matrix of carefully crafted structural properties. In that matrix, starting from the

---

[*]Tie-breaking rule when selecting between two or more different best-response actions.

[*]Due to the vastness of the literature in fictitious play, it is not possible to include all the works that have either been used or been inspired by this method.

lower-left element, the sequence of successive increments form a spiraling trajectory that converge towards the element of maximum value. We demonstrate that fictitious play has to follow the same trajectory, passing through all non-zero elements, to reach the unique pure Nash equilibrium, as illustrated in Figure 2. A crucial component of the proof is provided by an induction argument that *emulates* the movement of fictitious play and provides a super-exponential lower bound on the number of rounds needed.

## 2 Preliminaries

### 2.1 Notation and Definitions

**Notation** Let $\mathbb{R}$ be the set of real numbers, and $[n] = \{1, 2, \ldots, n\}$ be the set of actions. We define $\Delta_n$ as the probability simplex, which is the set of $n$-dimensional probability vectors, i.e., $\Delta_n := \{x \in \mathbb{R}^n : x_i \geq 0, \sum_{i=1}^{n} x_i = 1\}$. We use $e_i$ to denote the $i$-th elementary vector, and to refer to a coordinate of a vector, we use either $x_i$ or $[x]_i$. The superscripts are used to indicate the time at which a vector is referring to.

**Normal-form Games** In a *two-player normal-form* game we are given a pair of payoff matrices $A \in \mathbb{R}^{n \times m}$ and $B \in \mathbb{R}^{n \times m}$ where $n$ and $m$ are the respective pure strategies of the *row* and the *column* player. If the row player selects strategy $i \in [n]$, and the column player selects strategy $j \in [m]$, then their respective payoffs are $A_{ij}$ for the row player and $B_{ij}$ for the column player.

The agents can use randomization. A *mixed strategy* for the row player is a probability distribution $x \in \Delta_n$ over the $n$ rows, and a *mixed strategy* for the column player is a probability distribution $y \in \Delta_m$ over the $m$ columns. After the row player selects mixed strategy $x \in \Delta_n$, and the column player selects mixed strategy $y \in \Delta_m$, their expected payoffs are $x^\top A y$ and $x^\top B y$, respectively.

**Potential Games** A two-player potential game is a class of games that admit a unique function $\Phi$, referred to as a potential function, which captures the incentive of all players to modify their strategies. In other words, if a player deviates from their strategy, then the difference in payoffs is determined by a potential function $\Phi$ evaluated at those two strategy profiles. We can express this formally as follows:

**Definition 2.1** (Two-player Potential Game)**.** For any given pair of strategies $(x, y)$ and a pair of unilateral deviations $x'$ by the row player and $y'$ by the column player, the difference in their utility is equivalent to the difference in the potential function.

$$(x')^\top A y - x^\top A y = \Phi(x', y) - \Phi(x, y) \quad \text{and} \quad x^\top A(y') - x^\top A y = \Phi(x, y') - \Phi(x, y)$$

**Remark 2.2.** We note $\Phi$ is a function that characterizes the equilibria of the game as the strategy profiles that maximize the potential function.

In this work, we focus on a specific type of potential games called *identical payoff games*, where both players receive the same payoff.

**Definition 2.3** (Identical Payoff Games)**.** A two-player normal-form game $(A, B)$ is called *identical payoff* if and only if $A = B$.

In this scenario, it is apparent that the potential function is given by $\Phi(x, y) = x^\top A y$. Finally we provide the definition of an approximate NE.

**Definition 2.4** ($\epsilon$-Nash Equilibrium)**.** A strategy profile $(x^\star, y^\star) \in \Delta_n \times \Delta_m$ is called an $\epsilon$-approximate NE if and only if

$$(x^\star)^\top A y^\star \geq x^\top A y^\star - \epsilon \ \forall x \in \Delta_n \quad \text{and} \quad (x^\star)^\top B y^\star \geq (x^\star)^\top B y - \epsilon \ \forall y \in \Delta_m$$

In words, an approximate Nash equilibrium is a strategy profile in which no player can improve their payoff significantly by unilaterally changing their strategy, but the strategy profile may not necessarily satisfy the precise definition of a Nash equilibrium.

**Remark 2.5.** We highlight two special cases of the $\epsilon$-NE. Firstly, when $\epsilon$ is equal to zero, it is referred to as an *exact Nash equilibrium*. Secondly, when the support of strategies is of size 1, it is called a *pure Nash equilibrium*. It is worth noting that potential games always admit a pure NE.

## 2.2 Fictitious Play

Fictitious play is a natural uncoupled game dynamics at which each agent chooses a *best response* to their opponent's empirical mixed strategy. Since there might be several best response actions at a given round, fictitious play might contain different sequences of play; see Definition 2.6.

**Definition 2.6** (Fictitious Play)**.** An infinite sequence of pure strategy profiles $(i^{(1)}, j^{(1)}), \ldots, (i^{(t)}, j^{(t)}), \ldots$ is called a fictitious play sequence if and only if at each round $t \geq 2$,

$$i^{(t)} \in \operatorname*{argmax}_{i \in [n]} \sum_{s=1}^{t-1} A_{ij^{(s)}} \quad \text{and} \quad j^{(t)} \in \operatorname*{argmax}_{j \in [m]} \sum_{s=1}^{t-1} B_{i^{(s)}j} \tag{1}$$

The empirical strategy profile of row and column player at time $T$ is defined as $\hat{x}^{(T)} = \left(\frac{1}{T} \sum_{s=1}^{T} e_{i^{(s)}}\right)$ and $\hat{y}^{(T)} = \left(\frac{1}{T} \sum_{s=1}^{T} e_{j^{(s)}}\right)$ where $e_{i^{(t)}}, e_{j^{(t)}}$ are the elementary basis vectors.

**Definition 2.7** (Cumulative utility vector)**.** For an infinite sequence of pure strategy profiles $(i^{(1)}, j^{(1)}), \ldots, (i^{(t)}, j^{(t)}), \ldots$, the *cumulative utility* vectors of the row and column player at round $t \geq 1$ are defined as,

$$R^{(t)} = \sum_{s=1}^{t-1} Ae_{j^{(s)}} \quad \text{and} \quad C^{(t)} = \sum_{s=1}^{t-1} e_{i^{(s)}}^{\top} B.$$

**Remark 2.8.** Fictitious play assumes that each agent selects at each round $t \in [T]$ a strategy with *maximum cumulative utility*. The latter decision-making algorithm is also known as *Follow the Leader*. We remark that the latter alternative interpretation provides a direct generalization of fictitious play in $N$-player games.

In their seminal work, Monderer et al.[18] established that in case of identical payoff games the empirical strategies of any fictitious play sequence converges asymptotically to a NE.

**Theorem 2.9** ([18])**.** *Let a fictitious play sequence* $(i^{(1)}, j^{(1)}), \ldots, (i^{(t)}, j^{(t)}), \ldots$ *for an identical payoff game described with matrix* $A$. *Then, there exists a round* $T^{\star} \geq 1$ *such that for any* $t \geq T^{\star}$, *the empirical strategy profile* $(\hat{x}^{(t)}, \hat{y}^{(t)})$ *converges to a NE with a rate of* $1/t$.

On the positive side, Theorem 2.9 establishes that *any fictitious play sequence* converges to a Nash equilibrium in the case of potential games [*] On the negative side, Theorem 2.9 does not provide any convergence rates, since the round $T^{\star}$ depends on the specific fictitious play sequence and its dependence on the number of strategies is rather unclear.

## 3 Main Result

In this section, we outline the steps towards proving Theorem 3.1, as it is stated below. Firstly, we introduce a carefully constructed payoff matrix $A$ of size $n \times n$ and analyze its structural properties in Section 3.1. Next, in Section 3.2, we investigate the behavior of fictitious play when the game is an two-player identical payoff game with this matrix $A$. We also present a set of key statements that are necessary for proving the main theorem. Finally, in Section 3.3, we provide a proof for the fundamental Lemma 3.8.

**Theorem 3.1.** *Let an identical payoff game defined with the matrix* $A$ *of size* $n \times n$ *and consider any fictitious play sequence* $(i^{(1)}, j^{(1)}), \ldots, (i^{(t)}, j^{(t)}), \ldots$ *with* $(i^{(1)}, j^{(1)}) = (n, 1)$. *In case the empirical strategy profile* $(\hat{x}^{(T)}, \hat{y}^{(T)})$ *is an* $\epsilon$-*approximate Nash equilibrium then it holds*

$$T \geq \Omega \left( 4^n((n/2 - 2)!)^4 + \frac{1}{n\sqrt{\epsilon}} \right).$$

*Moreover, the lower bound on* $T$ *is independent of the tie-breaking rule. Finally, if the initialization is chosen uniformly at random, then the expected number of rounds to reach an* $\epsilon$-*approximate Nash equilibrium is* $\Omega \left( \frac{4^n((n/2-2)!)^4 + 1/n\sqrt{\epsilon}}{n^2} \right)$.

---

[*]For the sake of exposition, we have stated Theorem 2.9 only for the case of *identical payoff games*. However, we remark that the same theorem holds for general $N$-player potential games.

$$K^n(z) = \begin{pmatrix} (z+2) & 0 & \cdots & 0 & (z+3) \\ 0 & & & & 0 \\ \vdots & & K^{n-2}(z+4) & & 0 \\ 0 & & & & (z+4) \\ (z+1) & 0 & \cdots & 0 & 0 \end{pmatrix}$$

(a) Recursive construction of $A$.

$$K^6(0) = \begin{pmatrix} 2 & 0 & 0 & 0 & 0 & 3 \\ 0 & 6 & 0 & 0 & 7 & 0 \\ 0 & 0 & 10 & 11 & 0 & 0 \\ 0 & 0 & 9 & 0 & 8 & 0 \\ 0 & 5 & 0 & 0 & 0 & 4 \\ 1 & 0 & 0 & 0 & 0 & 0 \end{pmatrix}$$

(b) An example for $z = 0$ and $n = 6$.

## 3.1 Construction and Analysis of the Payoff Matrix $A$

We begin by introducing our recursive construction for the payoff matrix, which we use to establish the formal statement of Theorem 3.1.

**Definition 3.2.** For any $z > 0$ and $n$ even, consider the following $n \times n$ matrix $K^n(z)$.

1. For $n = 2$, $K^n(z) = \begin{pmatrix} z+2 & z+3 \\ z+1 & 0 \end{pmatrix}$.

2. For $n \geq 4$,

   - $K^n_{n1}(z) = z + 1$ and $K^n_{nj}(z) = 0$ for $j \notin \{1\}$.            *Row $n$*

   - $K^n_{n1}(z) = z + 1$, $K^n_{11}(z) = z + 2$ and $K^n_{1j}(z) = 0$ for any $j \notin \{1, n\}$.      *Column 1*

   - $K^n_{11}(z) = z + 2$, $K^n_{1n}(z) = z + 3$ and $K^n_{1j}(z) = 0$ for $j \notin \{1, n\}$.      *Row 1*

   - $K^n_{1n}(z) = z + 3$, $K^n_{n-1n}(z) = z + 4$ and $K^n_{nj}(z) = 0$ for $j \notin \{1, n - 1\}$.    *Column $n$*

   - For all $i, j \in \{2, \ldots, n - 1\} \times \{2, \ldots, n - 1\}$, $K^n_{ij}(z) := K^{n-2}_{i-1j-1}(z + 4)$.

In Figures 1a and 1b, we provide a schematic representation of Definition 3.2 and an illustrative example for $n = 6$ and $z = 0$. For the sake of simplicity, we have intentionally omitted the remaining zeros in the outer rows, and columns of the matrix.

The construction of the payoff matrix exhibits an interesting circular pattern, which begins at the lower left corner and extends along the outer layer of the matrix. More specifically, the first increment occurs in the same column as the starting point, i.e., at position $(1, 1)$ on the first row. The pattern then proceeds to the next greater element on the same row but a different column, i.e., at position $(1, n)$, and the last increment before entering the inner sub-matrix is located on the same column but on the $(n - 1)$-th row.

$$\text{Sequence of increments: } (n, 1) \rightarrow (1, 1) \rightarrow (1, n) \rightarrow (n - 1, n) \rightarrow \underbrace{(n - 1, 1) \rightarrow \cdots}_{K^{n-2}(z+4)}$$

The increments have been carefully selected to ensure that there are alternating changes in row and column when starting from the lower-left corner and following the successive increments, until reaching the sub-matrix in the center. Once inside the sub-matrix, a similar pattern continues. As we will explore later on, the structure of the payoff dictates the behavior of fictitious play. We denote the payoff matrix under consideration as $A$, which is defined as $A := K_n(0)$. The subsequent statements establish the key properties of $A$.

**Observation 3.3** (Structural Properties of matrix $A$). Let the matrix $A = K^n(0)$, then for $i \in \{0, \ldots, \frac{n}{2} - 1\}$ the following hold:

- The only elements with non-zero values in column $i + 1$ are located at positions $i + 1$ and $n - i$, and have values $4i + 2$ and $4i + 1$, respectively.

- The only non-zero elements of row $i + 1$ are located at positions $i + 1$ and $n - i$, and have values $4i + 2$ and $4i + 3$, respectively.

- The only non-zero elements of column $n - i$ are located at positions $i + 1$ and $n - i - 1$ and have values $4i + 3$ and $4i + 4$, respectively.

- The only non-zero elements of row $n - i$ are located at positions $i + 1$ and $n - i + 1$, and have values $4i + 1$ and $4i$, respectively.

**Observation 3.4.** The maximum value of $A$ is $2n - 2$ and is located at the entry $\left(\frac{n}{2}, \frac{n}{2} + 1\right)$.

**Proposition 3.5.** *For any non-zero element in the matrix $A$, there is at most one non-zero element that is greater and at most one non-zero element that is smaller in the same column or row.*

*Proof.* By Observation 3.3, each row and column of matrix $A$ contains at most two non-zero elements that are necessarily different to each other. Thus, if $(i, j)$ is a non-zero element of $A$, there can be at most two additional non-zero elements in row $i$ and column $j$ combined. Moreover, at most one of these elements can be greater and at most one can be smaller. $\square$

One of the central components of the main theorem is presented below. Lemma 3.6 establishes that in an identical payoff game with matrix $A$, any approximate Nash equilibrium must distribute the majority of its probability mass to the maximum element in $A$, which is located at the entry $\left(\frac{n}{2}, \frac{n}{2} + 1\right)$. The proof of this theorem is based solely on the structural properties presented in Observation 3.3, and is provided in the Appendix.

**Lemma 3.6** (Unique $\epsilon^2$-NE). *Let $\epsilon \in O\left(\frac{1}{n^3}\right)$ and consider an $\epsilon^2$-approximate Nash Equilibrium $(x^*, y^*)$. Then the following hold,*

$$x^*_{\frac{n}{2}} \geq 1 - n\epsilon \quad and \quad y^*_{\frac{n}{2}+1} \geq 1 - n\epsilon$$

Lemma 3.6 not only establishes that the only approximate Nash equilibrium of the identical-payoff game with matrix $A$ corresponds to the strategies $\left(\frac{n}{2}, \frac{n}{2} + 1\right)$, but it also implies that this is the only exact Nash equilibrium (i.e., $\epsilon = 0$). This observation follows from Observation 3.4, which states that the entry $\left(\frac{n}{2}, \frac{n}{2} + 1\right)$ corresponds to a maximum value of $A$ and hence it is a Nash equilibrium as it dominates both the row and the column that it belongs to. We can formally state this observation as follows.

**Corollary 3.7** (Unique pure NE). *In an identical-payoff game with payoff matrix $A$, there exists a unique pure Nash equilibrium at $\left(\frac{n}{2}, \frac{n}{2} + 1\right)$.*

### 3.2 Lower Bound for Fictitious Play in a Game with Matrix $A$

In this subsection, we present the proof of Theorem 3.1. To achieve this, we first prove that fictitious play requires super-exponential time before placing a positive amount of mass in entry $\left(\frac{n}{2}, \frac{n}{2} + 1\right)$. This result is established by our main technical contribution of the subsection, which is Lemma 3.8.

**Lemma 3.8.** *Let an identical-payoff game with payoff matrix $A$ and a fictitious play sequence $(i^{(1)}, j^{(1)}), \ldots, (i^{(t)}, j^{(t)}), \ldots$ with $(i^{(1)}, j^{(1)}) = (n, 1)$. Then, for all $\ell = \{0, \ldots, \frac{n}{2} - 1\}$ there exists a round $T_\ell \geq 1$ such that:*

1. *the agents play the strategies $(n - \ell, \ell + 1)$ for the first time,*

2. *all rows $r \in [\ell + 1, n - \ell - 1]$ admit $0$ cumulative utility, $R_r^{(T_\ell)} = 0$,*

3. *all columns $c \in [\ell + 2, n - \ell]$ admit $0$ cumulative utility, $C_c^{(T_\ell)} = 0$.*

*Moreover for $\ell \geq 2$, the cumulative utility of row $n - \ell$ at round $T_\ell$ is greater than*

$$R_{n-\ell}^{(T_\ell)} \geq 4\ell(4\ell - 1)(4\ell - 2)(4\ell - 3) \cdot R_{n-\ell}^{(T_{\ell-1})} \quad while \quad R_{n-1}^{(T_1)} \geq 4. \tag{2}$$

Using Lemma 3.8 we are able to establish that for a very long period of time the row player has never played row $\frac{n}{2}$ and that the column player has never played column $\frac{n}{2} + 1$.

**Lemma 3.9** (Exponential Lower Bound). *Let an identical-payoff game with matrix $A$ and a ficti­tious play sequence $(i^{(1)}, j^{(1)}), \ldots, (i^{(t)}, j^{(t)}), \ldots$ with $(i^{(1)}, j^{(1)}) = (n, 1)$. In case $(i^{(T)}, j^{(T)}) = \left(\frac{n}{2}, \frac{n}{2} + 1\right)$ then $T \geq \Omega(4^n((n/2 - 2)!)^4)$.*

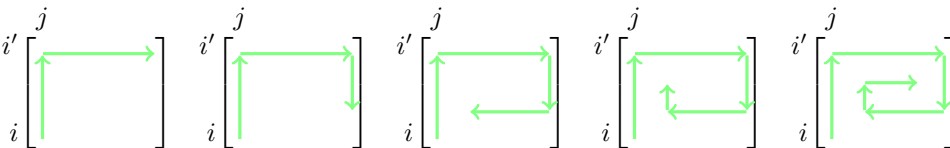

Figure 2: This figure shows the spiral trajectory generated by the fictitious play dynamic in a game with matrix $A$. The starting point is the lower-left element, and as the dynamic progresses, it visits all non-zero elements in ascending order of value.

*Proof.* Based on Lemma 3.8, we can guarantee the existence of a round $T^\star := T_{n/2-1}$ when the players choose the strategy profile $(\frac{n}{2}+1, \frac{n}{2})$ for the first time. In addition, at round $T^\star$, it holds that $R_{n/2-1}^{(T^\star)} > 0$ and $C_{n/2+1}^{(T^\star)} = R_{n/2}^{(T^\star)} = 0$. The latter condition ensures that the strategy profile $(\frac{n}{2}, \frac{n}{2}+1)$ has not been played up to time $T^\star$.

As indicated by Observation 3.3, row $\frac{n}{2}$ has non-zero entries at columns $\frac{n}{2}$ and $\frac{n}{2}+1$. Therefore, if the cumulative utilities $R_{n/2}^{(T^\star)}$ at time $T^\star$ is zero, this implies that neither of these columns has been chosen up to that point. By the same reasoning, column $\frac{n}{2}+1$ has a non-zero entry only at row $\frac{n}{2}$, indicating that this row has not been chosen as well.

In order to continue, we require an estimate of the duration during which the strategy profile $(\frac{n}{2}+1, \frac{n}{2})$ will be played. Observation 3.3 guarantees that the utility vector of the row player is the following.

$$A e_{\frac{n}{2}} = (0, \ldots, 0, \underbrace{2n-2}_{\frac{n}{2}}, \underbrace{2n-3}_{\frac{n}{2}+1}, 0, \ldots, 0) \tag{3}$$

We can combine this information with the fact that $R_{n/2+1}^{(T^\star)} \neq 0$, $R_{n/2}^{(T^\star)} = 0$, and that the respective entries in the payoff vector (3) of the row player differ by exactly one. This allows us to conclude that it will take at least $R_{\frac{n}{2}+1}^{(T^\star)}$ iterations for the cumulative utilities to become equal, i.e $R_{\frac{n}{2}+1} = R_{\frac{n}{2}}$. Therefore, the lower bound on the number of iterations holds regardless of the tie-breaking rule.

Now, if at any time $T$ the agents play the strategy profile $(i^{(T)}, j^{(T)}) = (\frac{n}{2}, \frac{n}{2}+1)$, we can conclude that $T \geq R_{\frac{n}{2}+1}^{(T^\star)}$. Using Equation (2) of Lemma 3.8 we obtain:

$$T \geq R_{\frac{n}{2}-1}^{(T^\star)} \geq 16^{\frac{n}{2}-1} \left( \prod_{\ell=2}^{\frac{n}{2}-1} (\ell-1) \right)^4 R_{n-1}^{(T_1)} \geq \Omega\left( 4^n \left( \left(\frac{n}{2}-2\right)! \right)^4 \right)$$

$\square$

Now that we have established the necessary technical result in Lemma 3.9, we are ready to present the proof of Theorem 3.1.

*Proof of Theorem 3.1.* Let $T^\star \geq 1$ denote the first time at which $(i^{(t)}, j^{(t)}) = (\frac{n}{2}, \frac{n}{2}+1)$. For any $t \leq T^\star - 1$, it holds that $\hat{x}_{\frac{n}{2}}^{(t)} = \hat{y}_{\frac{n}{2}+1}^{(t)} = 0$. Thus, Lemma 3.6 implies that $(\hat{x}^{(t)}, \hat{y}^{(t)})$ is not an approximate NE. On the other hand, at each round $t \geq T^\star$ we are ensured that $\hat{x}_{\frac{n}{2}}^{(t)}, \hat{y}_{\frac{n}{2}+1}^{(t)}$ converges to 1 with rate $1/t$. Applying Lemma 3.6 for $\epsilon := \sqrt{\epsilon}$ we get that if $(\hat{x}^{(t)}, \hat{y}^{(t)})$ is an $\epsilon$-NE then $t \geq T^\star + \frac{1}{n\sqrt{\epsilon}} \geq \Omega\left(4^n((n/2-2)!)^4\right) + \frac{1}{n\sqrt{\epsilon}}$. It is evident that, even with a uniformly random initialization, the probability of selecting $(n, 1)$ as the starting point for fictitious play is $1/n^2$ and so the claim of the theorem follows. $\square$

$$
\begin{pmatrix} 4i+2 & 4i+3 \\ & 4i+4 \\ 4i+1 & & 4i \end{pmatrix}
\quad
\begin{pmatrix} 4i+2 & 4i+3 \\ & 4i+4 \\ 4i+1 & & 4i \end{pmatrix}
\quad
\begin{pmatrix} 4i+2 & 4i+3 \\ & 4i+4 \\ 4i+1 & & 4i \end{pmatrix}
\quad
\begin{pmatrix} 4i+2 & 4i+3 \\ & 4i+4 \\ 4i+1 & & 4i \end{pmatrix}
$$

(a) $t \in [T_i^0, T_i^1)$     (b) $t \in [T_i^1, T_i^2)$     (c) $t \in [T_i^2, T_i^3)$     (d) $t \in [T_i^3, T_i^4)$

Figure 3: The figure illustrates the active row and column player for each time period, with the played strategy highlighted in purple and the corresponding payoff vectors of the row and column player highlighted in green. Additionally, the time period of each played strategy is indicated for clarity.

### 3.3 Strategy Switches and Proof of Lemma 3.8

This subsection is dedicated to presenting the proof of Lemma 3.8. However, before delving into the proof, we first provide additional statements that serve to shed light on the sequence of strategy profiles generated by fictitious play.

According to Proposition 3.5, if the strategy $(i, j)$ is chosen by fictitious play, then either in row $i$ or in column $j$, there is at most one element greater than $A_{ij}$. Therefore, in a subsequent round, fictitious play will necessarily choose this element as its strategy. Let $i'$ be the row where the element of greater value is located. Intuitively, we can imagine that as fictitious play continues to play the same strategy $(i, j)$, the payoff vector $Ae_j$ is repeatedly added to the cumulative vector of the row player. Since this vector has a greater value at coordinate $i'$ than at coordinate $i$, a strategy switch will eventually occur. We state this observation in Proposition 3.10 and defer the proof to the Appendix.

**Proposition 3.10** (Strategy Switch). *Let $(i^{(t)}, j^{(t)})$ be a strategy selected by fictitious play at round $t$, and $(i^{(t)}, j^{(t)}) \neq (\frac{n}{2}, \frac{n}{2})$. Then, in a subsequent round, fictitious play will choose the strategy of greater value that is either on row $i^{(t)}$ or column $j^{(t)}$.*

As previously hinted, the structure of the payoff matrix dictates the sequence of strategy profiles chosen by fictitious play. The sequence of strategy switches alternates between rows and columns. Continuing with the example from the previous paragraph, let us assume that the row player made the most recent strategy switch from $(i, j)$ to $(i', j)$. This implies that the element $A_{i'j}$ is greater than $A_{ij}$, as otherwise a strategy switch would not have taken place, as established in Proposition 3.10. Moreover, by Proposition 3.5, we know that there must be an element of greater value in either row $i'$ or column $j$. Since $A_{i'j}$ is greater than $A_{ij}$, any element of greater value must be located in row $i'$. The same reasoning applies for the case where the column player was the last to switch strategies. We can summarize this observation by stating that if one player is the last to switch, then the other player must switch next. We formally state this in Corollary 3.11 and defer the proof to the Appendix.

**Corollary 3.11** (Successive Strategy Switches). *Let $t$ be a round in which a player changes their strategy. Then exactly one of the following statements is true:*

1. *If the row player changes their strategy at round $t$, i.e. $i^{(t)} \neq i^{(t-1)}$, then the column player can only make the next strategy switch.*

2. *If the column player changes their strategy at round $t$, i.e. $j^{(t)} \neq j^{(t-1)}$, then the row player can only make the next strategy switch.*

Applying the same concept, we can observe that starting from the lower-left corner, fictitious play follows a spiral trajectory. The resulting spiral is illustrated in Figure 2. We now proceed with the main result of this section, Lemma 3.8.

*Proof.* Since $(i^{(1)}, j^{(1)}) = (n, 1)$ all the above claims trivially for $T_0 = 1$. We assume that the claim holds for $i$ and will now establish it inductively for $i + 1$.

By the induction hypothesis, agents play strategies $(n - i, i + 1)$ at round $T_i^0 := T_i$. Furthermore, row $n - i$ admits cumulative utility of $R_{n-i}^{(T_i^0)}$ while row $i + 1$ admits cumulative utility of $R_{i+1}^{(T_i^0)} = 0$. According to Observation 3.3, the payoff vectors of the row and column agent are highlighted in Figure 3a. By combining these facts, we can establish the following.

**Proposition 3.12** (Abridged; Full Version in Proposition B.3). *There exists a round $T_i^1 > T_i^0$ at which the strategy profile is $(i + 1, i + 1)$ for the first time, column $i + 1$ admits cumulative utility $C_{i+1}^{(T_i^1)} \geq (4i + 1) \cdot (R_{i+1}^{(T_i^0)} + 1)$, and $C_{n-i}^{(T_i^1)} = 0$.*

By Proposition 3.12, at round $T_i^1$, the agents play strategies $(i+1, i+1)$. Furthermore, the cumulative utility of column $n - i$ equals to $C_{n-i}^{(T_i^1)} = 0$. According to Observation 3.3, the payoff vectors of the row and column agent are highlighted Figure 3b. Combining these facts, we get the following:

**Proposition 3.13** (Abridged; Full Version in Proposition B.4). *There exists a round $T_i^2 > T_i^1$ at which the strategy profile is $(i + 1, n - i)$ for the first time, row $i + 1$ admits cumulative utility $R_{i+1}^{(T_i^2)} \geq (4i + 2) \cdot C_{i+1}^{(T_i^1)}$, and $R_{n-i-1}^{(T_i^2)} = 0$.*

By Proposition 3.13, at round $T_i^2$, the agents play strategies $(i+1, n-i)$. Furthermore, the cumulative utility of row $n - i - 1$ equals to $R_{n-i-1}^{(T_i^2)} = 0$. According to Observation 3.3, the payoff vectors of the row and column agent are highlighted Figure 3c. By combining these facts, we get the following:

**Proposition 3.14** (Abridged; Full Version in Proposition B.5). *There exists a round $T_i^3 > T_i^2$ at which the strategy profile is $(n - (i + 1), n - i)$ for the first time, $R_{i+2}^{(T_i^3)} = 0$.*

By Proposition 3.14, at round $T_i^3$, the agents play strategies $(n - i - 1, n - i)$. the cumulative of column $i + 2$ equals to $R_{i+2}^{(T_i^3)} = 0$. According to Observation 3.3, the payoff vectors of the row and column agent are highlighted Figure 3d. By combining these facts, we can establish the following.

**Proposition 3.15** (Abridged; Full Version in Proposition B.6). *There exists a round $T_i^4 > T_i^3$ at which the strategy profile is $(n-(i+1), (i+1)+1)$ for the first time, row $n-i-1$ admits cumulative utility $R_{n-(i+1)}^{(T_i^4)} \geq (4i + 4) \cdot C_{n-i}^{(T_i^3)}$, all rows $k \in [(i + 1) + 1, n - (i + 1) - 1]$ admit $R_k^{(T_i^4)} = 0$ and all columns $k \in [(i + 1) + 2, n - (i + 1)]$ admit $C_k^{(T_i^4)} = 0$.*

Proposition 3.15 establishes that there exits a round $T_{i+1} := T_i^4$ at which the strategy profile $(n - (i + 1), (i + 1) + 1)$ is played for the first time. Furthermore, Proposition 3.15 confirms that all rows $k \in \{(i+1)+1, n-(i+1)-1\}$ admit $R_k^{(T_{i+1})} = 0$ and all columns $k \in \{(i+1)+2, n-(i+1)\}$ admit $C_k^{(T_{i+1})} = 0$. We still need to verify the the recursive relation Equation (2). By combining Proposition 3.12, 3.13, 3.14 and 3.15, we can deduce

$$R_{n-i-1}^{(T_{i+1})} \geq (4i + 4)(4i + 3)(4i + 2)(4i + 1)R_{n-i}^{(T_i)}.$$

$\square$

## 4 Experiments

In this section, we aim to experimentally validate our findings on a $4 \times 4$ payoff matrix. Our analysis focuses on three key aspects: the round in which a new strategy switch occurs, the Nash gap throughout the game, and the empirical strategy employed by the $x$ player. We present the plot from the row player's perspective, which is identical to that of the column player.

In Figure 4a, we provide the time steps of all strategy switches. As it is expected from the analysis, fictitious play *visits* all strategies, specifically in increasing order of their utility, to reach the pure Nash equilibrium. Moreover, in Figure 4b we observe a recurring pattern in the Nash gap diagram, where the gap increases after the selection of a new strategy with a higher utility and decreases until the next strategy switch. However, this pattern stops after the pure Nash equilibrium is reached, which is the unique approximate Nash equilibrium in accordance with Lemma 3.8.

| $t$ | $player$ | $from$ | $to$ |
|-----|----------|--------|------|
| 2   | $i$      | 4      | 1    |
| 4   | $j$      | 1      | 4    |
| 11  | $i$      | 1      | 3    |
| 38  | $j$      | 4      | 2    |
| 174 | $i$      | 3      | 2    |
| 990 | $j$      | 2      | 3    |

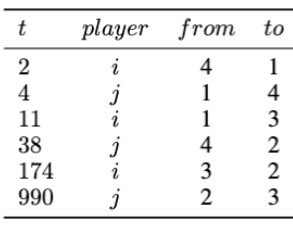 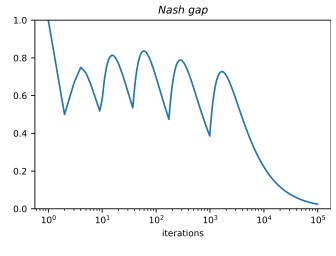 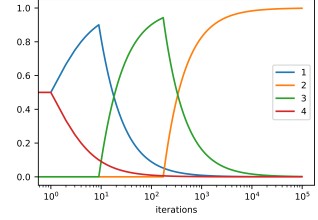

(a) Table of transitions      (b) Nash Gap      (c) Empirical strategy of $x$ player

Figure 4: The figure displays the three aspects that were analyzed in the experiments. To preserve the crucial qualitative features in both diagrams, the $x$-axis was set to a logarithmic scale.

## 5 Conclusion

In summary, this paper has provided a thorough examination of the convergence rate of fictitious play within a specific subset of potential games. Our research has yielded a recursive rule for constructing payoff matrices, demonstrating that fictitious play, regardless of the tie-breaking rule employed, may require super exponential time to reach a Nash equilibrium even in two-player identical payoff games. This contribution to the literature differs from previous studies and sheds new light on the limitations of fictitious play in the particular class of potential games.

**Limitations and Broader Impacts:** Our work is theoretical in nature, and we do not identify any limitations or negative ethical or societal implications.

## Acknowledgments

Ioannis Panageas would like to acknowledge a start-up grant and SF50818. This work was supported by the Swiss National Science Foundation (SNSF) under grant number 200021_205011, by Hasler Foundation Program: Hasler Responsible AI (project number 21043) and Innovation project supported by Innosuisse (contract agreement 100.960 IP-ICT).

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

# Appendix

## A  Proof of Lemma 3.6

In this section, we present a comprehensive proof of Lemma 3.6. This lemma establishes a fundamental property of our recursive construction, as defined in Definition 3.2: the only $\epsilon^2$-approximate Nash equilibrium is located at the maximum value element. Consequently, the fictitious play dynamics fail to reach even an approximate Nash equilibrium (within the specified degree of approximation) unless the strategy $\left(\frac{n}{2}, \frac{n}{2}+1\right)$ has been played for a sufficient duration. Our theoretical findings are further supported by our experiments. As shown in Figure 4b, the Nash gap does not vanish before the final strategy switch occurs. To provide a precise definition of the Nash gap, we present it below.

**Definition A.1** (Nash Gap for Identical Payoff). The Nash gap at round $t$ for a two-player identical payoff games is

$$\left(\max_{i\in[n]}[Ay^{(t)}]_i - (x^{(t)})^\top Ay^{(t)}\right) + \left(\max_{j\in[n]}[(x^{(t)})^\top A]_j - (x^{(t)})^\top Ay^{(t)}\right)$$

To establish Lemma 3.6, we employ a similar approach as in the other proofs presented in this work. Specifically, we heavily rely on the structure of our payoff matrix and utilize an induction technique to demonstrate that the majority of the probability mass must be concentrated in the maximum element. The induction argument starts from the outermost elements of the matrix, namely row $n-1$, column 1, and row 1, column $n-1$. By successive induction steps, we establish that until the maximum element is reached, none of the elements in those rows or columns can possess a significant probability mass.

**Lemma A.2** (Proof of Lemma 3.6). *Let $\epsilon \in (0, 1/56n^3]$ and $(x^\star, y^\star)$ an $\epsilon^2$-NE. Then for each $i \in [0, n/2 - 2]$,*

- *$x^\star_{i+1} \leq \epsilon$ and $x^\star_{n-i} \leq \epsilon$.*
- *$y^\star_{i+1} \leq \epsilon$ and $y^\star_{n-i} \leq \epsilon$.*

*Proof.* We will prove the claim by induction. First, assume that for all $j \in [0, i-1]$, we have:

- $x^\star_{j+1} \leq \epsilon$ and $x^\star_{n-j} \leq \epsilon$.
- $y^\star_{j+1} \leq \epsilon$ and $y^\star_{n-j} \leq \epsilon$.

Next, we proceed to establish the inequalities $x^\star_{n-i}, y^\star_{i+1}, x^\star_{i+1}, y^\star_{n-i} \leq \epsilon$. We demonstrate these inequalities in the exact order as presented, as their proof relies on the underlying structure of the matrix. It is important to note that there are interdependencies between these inequalities, which we will address accordingly.

**Case 1:** $x^\star_{n-i} \leq \epsilon$

Let assume that $x^\star_{n-i} > \epsilon$ and we will reach a contradiction. From Observation 3.3, we notice that the utility of row $n-i$ equals

$$[Ay^\star]_{n-i} = (4i+1)y^\star_{i+1} + 4iy^\star_{n-i+1} \tag{4}$$

At the same time the utility of row $i+1$ equals

$$[Ay^\star]_{i+1} = (4i+2)y^\star_{i+1} + (4i+3)y^\star_{n-i}$$

As a result, by taking the difference on the utilities of row $i+1$ and $n-i$ we get,

$$
\begin{aligned}
[Ay^\star]_{i+1} - [Ay^\star]_{n-i} &= (4i+2)y^\star_{i+1} + (4i+3)y^\star_{n-i} - (4i+1)y^\star_{i+1} - 4iy^\star_{n-i+1} \\
&= y^\star_{i+1} + (4i+3)y^\star_{n-i} - 4iy^\star_{n-i+1} \\
&\geq y^\star_{i+1} + y^\star_{n-i} - 4i\epsilon
\end{aligned}
$$

where the last inequality follows by the fact that $y^\star_{n-i+1} \leq \epsilon$ (Inductive Hypothesis). As a result, we conclude that

$$[Ay^\star]_{i+1} - [Ay^\star]_{n-i} \geq y_{i+1}^\star + y_{i+1}^\star - 4i\epsilon$$

In case $y_{i+1}^\star + y_{n-i}^\star \geq (4i+1)\epsilon$ then $[Ay^\star]_{i+1} - [Ay^\star]_{n-i} \geq \epsilon$. Hence if the row player puts $x_{n-i}^\star$ probability mass to row $i+1$ by transferring the probability mass from row $n-i$ to row $i+1$ then it increases its payoff by $x_{n-i}^\star([Ay^\star]_{i+1} - [Ay^\star]_{n-i}) > \epsilon^2$. The latter contradicts with the assumption that $(x^\star, y^\star)$ is an $\epsilon^2$-NE. Thus, we conclude in the following two statements,

$$y_{i+1}^\star + y_{n-i}^\star \leq (4i+1)\epsilon \quad \text{and} \quad [Ay^\star]_{n-i} \leq 2(4i+1)^2\epsilon$$

where the last inequality is obtained by combining the first inequality with Equation (4). Now consider the sum of the utilities of rows $k \in [i+2, n-i-1]$. By the construction of the payoff matrix $A$ we can easily establish the following claim.

**Proposition A.3.** *The sum of utilities of rows $k \in [i+2, n-i-1]$ satisfies the inequality,*

$$\sum_{k=i+2}^{n-i-1} [Ay^\star]_k \geq \sum_{k=i+2}^{n-i-1} y_k^\star$$

By Proposition A.3 we are ensured that

$$
\begin{aligned}
\sum_{k=i+2}^{n-i-1} [Ay^\star]_k &\geq \sum_{k=i+2}^{n-i-1} y_k^\star \\
&= 1 - \sum_{k=1}^{i+1} y_k^\star - \sum_{k=n-i}^{n} y_k^\star \\
&= 1 - (y_{i+1}^\star + y_{n-i}^\star) - \left( \sum_{k=1}^{i} y_k^\star + \sum_{k=n-i+1}^{n} y_k^\star \right) \\
&\geq 1 - (4i+1)\epsilon - n\epsilon \\
&\geq 1 - (5n+1)\epsilon
\end{aligned}
$$

where the second inequality follows by the fact that $y_{i+1}^\star + y_{n-i}^\star \leq (4i+1)\epsilon$ and the fact that $y_j^\star \leq \epsilon$ for all $k \in [1, i] \cup [n-i+1, n]$ (Inductive Hypothesis).

Due to the fact that

$$\sum_{k=i+2}^{n-i-1} [Ay^\star]_k \geq 1 - (5n+1)\epsilon$$

we are ensured that there exists a row $k^\star \in [i+2, n-i-1]$ with utility $[Ay^\star]_{k^\star} \geq \frac{1-(5n+1)\epsilon}{n}$. Now consider the difference between the utility of row $k^\star$ and the row $n-i$.

$$[Ay^\star]_{k^\star} - [Ay^\star]_{n-i} \geq \frac{1-(5n+1)\epsilon}{n} - 2(4i+1)^2\epsilon \geq \frac{1-(5n+1)\epsilon}{n} - 2(4n+1)^2\epsilon \geq \epsilon$$

where the last inequality holds for $\epsilon \leq 1/56n^3$. Hence if the row player puts $x_{n-i}^\star$ probability mass to row $k^\star$ then it increases its payoff by $x_{n-i}^\star([Ay^\star]_{k^\star} - [Ay^\star]_{n-i}) > \epsilon^2$. The latter contradicts with the assumption that $(x^\star, y^\star)$ is an $\epsilon^2$-NE. Thus we have reached to a final contradiction that $x_{n-i}^\star > \epsilon$.

**Case 2:** $y_{i+1}^\star \leq \epsilon$

Similar to the previous case, we assume that $y_{i+1}^\star > \epsilon$ and proceed to derive a contradiction. From Observation 3.3, we notice that the utility of column $i+1$ is given by:

$$[(x^\star)^\top A]_{i+1} = (4i+2)x_{i+1}^\star + (4i+1)x_{n-i}^\star$$

At the same time the utility of column $n - i$ equals

$$[(x^\star)^\top A]_{n-i} = (4i + 3)x^\star_{i+1} + (4i + 4)x^\star_{n-i-1}$$

As a result, by taking the difference on the utilities of columns $n - i$ and $i + 1$ we get,

$$
\begin{aligned}
[(x^\star)^\top A]_{n-i} - [(x^\star)^\top A]_{i+1} &= (4i + 3)x^\star_{i+1} + (4i + 4)x^\star_{n-i-1} - (4i + 2)x^\star_{i+1} - (4i + 1)x^\star_{n-i} \\
&= x^\star_{i+1} + (4i + 4)x^\star_{n-i-1} - (4i + 1)x^\star_{n-i} \\
&\geq x^\star_{i+1} + x^\star_{n-i-1} - (4i + 1)\epsilon
\end{aligned}
$$

where the last inequality follows by the fact $x^\star_{n-i} \leq \epsilon$ (Inductive Step Case 1). As a result, we conclude that

$$[(x^\star)^\top A]_{n-i} - [(x^\star)^\top A]_{i+1} \geq x^\star_{i+1} + x^\star_{n-i-1} - (4i + 1)\epsilon$$

In case $x^\star_{i+1} + x^\star_{n-i-1} \geq (4i + 2)\epsilon$ then $[(x^\star)^\top A]_{n-i} - [(x^\star)^\top A]_{i+1} \geq \epsilon$. Hence if the column player puts $y^\star_{i+1}$ probability mass to column $n - i$ then it increases its payoff by $y^\star_{i+1}([(x^\star)^\top A]_{n-i} - [(x^\star)^\top A]_{i+1}) > \epsilon^2$. The latter contradicts with the assumption that $(x^\star, y^\star)$ is an $\epsilon^2$-NE. Thus we conclude in the following two statements:

$$x^\star_{i+1} + x^\star_{n-i-1} \leq (4i + 2)\epsilon \quad \text{and} \quad [(x^\star)^\top A]_{i+1} \leq 2(4i + 2)^2\epsilon$$

Now consider the sum of the utilities of columns $k \in [i + 2, n - i - 1]$. By the construction of the payoff matrix $A$ we can easily establish the following claim.

**Proposition A.4.** *The sum of utilities of columns $k \in [i + 2, n - i - 1]$ satisfies the inequality,*

$$\sum_{k=i+2}^{n-i-1} [(x^\star)^\top A]_k \geq \sum_{k=i+2}^{n-i-1} x^\star_k$$

By Proposition A.4 we are ensured that

$$
\begin{aligned}
\sum_{k=i+2}^{n-i-1} [(x^\star)^\top A]_k &\geq \sum_{k=i+2}^{n-i-1} x^\star_k \\
&= 1 - \sum_{k=1}^{i+1} x^\star_j - \sum_{k=n-i}^{n} x^\star_j \\
&= 1 - (x^\star_{i+1} + x^\star_{n-i}) - \left( \sum_{k=1}^{i} x^\star_i + \sum_{k=n-i+1}^{n} x^\star_j \right) \\
&\geq 1 - ((4i + 2)\epsilon + \epsilon) - \left( \sum_{k=1}^{i} x_i + \sum_{k=n-i+1}^{n} x^\star_j \right) \\
&\geq 1 - (5n + 3)\epsilon
\end{aligned}
$$

where the second to last inequality follows by the facts: $x^\star_{i+1} + x^\star_{n-i-1} \leq (4i + 2)\epsilon$ and so $x^\star_{i+1} \leq (4i + 2)\epsilon$, and the Inductive Step Case 1, $x^\star_{n-i} \leq \epsilon$. Moreover, the last inequality holds by the Inductive Hypothesis: $x^\star_k \leq \epsilon$ for all $k \in [1, i] \cup [n - i + 1, n]$.

Due to the fact that

$$\sum_{k=i+2}^{n-i-1} [(x^\star)^\top A]_k \geq 1 - (5n + 3)\epsilon$$

we are ensured that there exists a column $k^\star \in [i + 2, n - i - 1]$ with utility $[(x^\star)^\top A]_{k^\star} \geq \frac{1 - (5n+3)\epsilon}{n}$. Now consider the difference between the utility of column $k^\star$ and the column $i + 1$.

$$[(x^\star)^\top A]_{k^\star} - [(x^\star)^\top A]_{i+1} \geq \frac{1 - (5n+3)\epsilon}{n} - 2(4i+2)^2\epsilon \geq \frac{1 - (5n+3)\epsilon}{n} - 2(4n+2)^2\epsilon \geq \epsilon$$

where the last inequality follows by the fact that $\epsilon \leq 1/56n^3$. Hence if the column player puts $y^\star_{i+1}$ probability mass to column $k^\star$ then it increases its payoff by $y^\star_{i+1}([(x^\star)^\top A]_{k^\star} - [(x^\star)^\top A]_{i+1}) > \epsilon^2$. The latter contradicts with the assumption that $(x^\star, y^\star)$ is an $\epsilon^2$-NE. Thus we have reached to a final contradiction that $y^\star_{i+1} > \epsilon$.

**Case 3:** $x^\star_{i+1} \leq \epsilon$

Let assume that $x^\star_{i+1} > \epsilon$ and we will reach a contradiction. From Observation 3.3, we notice that the utility of row $i+1$ equals,

$$[Ay^\star]_{i+1} = (4i+2)y^\star_{i+1} + (4i+3)y^\star_{n-i}$$

Next, we examine the row $n-(i+1)$ in the inner submatrix. If this row is not well-defined, it implies that the inductive step $j = i$ has reached the $2 \times 2$ submatrix. In this case, the proof of Lemma 3.6 has already been completed. Otherwise, the utility of row $n-(i+1)$ equals

$$[Ay^\star]_{n-(i+1)} = (4(i+1)+1)y^\star_{(i+1)+1} + (4(i+1))y^\star_{n-(i+1)+1} \geq (4i+4)y^\star_{n-i}$$

As a result, by taking the difference on the utilities of row $i+1$ and $n-(i+1)$ we get,

$$
\begin{aligned}
[Ay^\star]_{n-(i+1)} - [Ay^\star]_{i+1} &= (4i+4)y^\star_{n-i} - (4i+2)y^\star_{i+1} - (4i+3)y^\star_{n-i} \\
&= y^\star_{n-i} - (4i+2)y^\star_{i+1} \\
&\geq y^\star_{n-i} - (4i+2)\epsilon
\end{aligned}
$$

where the last inequality follows by the fact that $y^\star_{i+1} \leq \epsilon$ (Inductive Step Case 2). As a result, we conclude that

$$[Ay^\star]_{n-(i+1)} - [Ay^\star]_{i+1} \geq y^\star_{n-1} - (4i+2)\epsilon$$

In case $y^\star_{n-i} \geq (4i+3)\epsilon$ then $[Ay^\star]_{n-(i+1)} - [Ay^\star]_{i+1} \geq \epsilon$. Hence, if the row player puts $x^\star_{i+1}$ probability mass to row $n-(i+1)$ then it increases its payoff by $x^\star_{i+1}([Ay^\star]_{n-(i+1)} - [Ay^\star]_{i+1}) > \epsilon^2$. The latter contradicts with the assumption that $(x^\star, y^\star)$ is an $\epsilon^2$-NE. Thus, we conclude the following two statements:

$$y^\star_{n-i} \leq (4i+3)\epsilon \quad \text{and} \quad [Ay^\star]_{i+1} \leq 2(4i+3)^2\epsilon$$

Now consider the sum of the utilities of rows $k \in [i+2, n-i-1]$. From Proposition A.3

$$
\begin{aligned}
\sum_{k=i+2}^{n-i-1} [Ay^\star]_k &= \sum_{k=i+2}^{n-i-1} y^\star_k = 1 - \left(\sum_{k=1}^{i+1} y^\star_k + \sum_{k=n-i}^{n} y^\star_k\right) \geq 1 - y^\star_{n-i} - \left(\sum_{k=1}^{i+1} y^\star_k + \sum_{k=n-i+1}^{n} y^\star_k\right) \\
&\geq 1 - (4i+3)\epsilon - n\epsilon = 1 - (5n+3)\epsilon
\end{aligned}
$$

Thus, we are ensured that there exists a row $k^\star \in [i+2, n-i-1]$ with utility $[Ay^\star]_{k^\star} \geq \frac{1-(5n+3)\epsilon}{n}$. Now consider the difference between the utility of row $k^\star$ and the row $n-i$.

$$[Ay^\star]_{k^\star} - [Ay^\star]_{i+1} \geq \frac{1 - (5n+3)\epsilon}{n} - 2(4i+3)^2\epsilon \geq \frac{1 - (5n+3)\epsilon}{n} - 2(4n+3)^2\epsilon \geq \epsilon$$

where the last inequality follows by the fact that $\epsilon \leq 1/56n^3$. Hence if the row player puts $x^\star_{i+1}$ probability mass to row $k^\star$ then it increases its payoff by $x^\star_{i+1}([Ay^\star]_{k^\star} - [Ay^\star]_{i+1}) > \epsilon^2$. The

latter contradicts with the assumption that $(x^\star, y^\star)$ is an $\epsilon^2$-NE. Thus we have reached to a final contradiction that $x^\star_{i+1} > \epsilon$.

**Case 4:** $y^\star_{n-i} \leq \epsilon$

Let assume that $y^\star_{n-i} > \epsilon$ and we will reach a contradiction. From Observation 3.3, we notice that the utility of column $n - i$ equals,

$$[(x^\star)^\top A]_{n-i} = (4i + 3)x^\star_{i+1} + (4i + 4)x^\star_{n-(i+1)}$$

Now, we consider the column $(i + 1) + 1$ in the inner submatrix. In case this column is not well-defined, it means that the inductive step $j = i$ has reached the $2 \times 2$ submatrix and so the proof of the Lemma 3.6 has already been completed. Otherwise, the utility of column $(i + 1) + 1$ equals

$$[(x^\star)^\top A]_{(i+1)+1} = (4(i + 1) + 2)x^\star_{(i+1)+1} + (4(i + 1) + 1)x^\star_{n-(i+1)+1} \geq (4i + 5)x^\star_{n-i}$$

As a result, by taking the difference on the utilities of columns $i + 1$ and $n - (i + 1)$ we get,

$$
\begin{aligned}
[(x^\star)^\top A]_{(i+1)+1} - [(x^\star)^\top A]_{n-i} &= (4i + 5)x^\star_{n-i} - (4i + 3)x^\star_{i+1} - (4i + 4)x^\star_{n-(i+1)} \\
&= x^\star_{n-i} - (4i + 3)x^\star_{i+1} \\
&\geq x^\star_{n-i} - (4i + 3)\epsilon
\end{aligned}
$$

where the last inequality follows by the fact that $x^\star_{i+1} \leq \epsilon$ (Inductive Step Case 3). As a result, we conclude that

$$[(x^\star)^\top A]_{(i+1)+1} - [(x^\star)^\top A]_{n-i} \geq x^\star_{n-i} - (4i + 3)\epsilon$$

In case $x^\star_{n-i} \geq (4i+4)\epsilon$ then $[(x^\star)^\top A]_{(i+1)+1} - [(x^\star)^\top A]_{n-i} \geq \epsilon$. Hence if the column player puts $y^\star_{n-i}$ probability mass to column $(i + 1) + 1$ then it increases its payoff by $y^\star_{n-i}([(x^\star)^\top A]_{(i+1)+1} - [(x^\star)^\top A]_{n-i}) > \epsilon^2$. The latter contradicts with the assumption that $(x^\star, y^\star)$ is an $\epsilon^2$-NE. Thus we conclude in the following two statements:

$$x^\star_{n-i} \leq (4i + 4)\epsilon \quad \text{and} \quad [(x^\star)^T A]_{n-i} \leq 2(4i + 4)^2\epsilon$$

Now consider the sum of the utilities of columns $k \in [i + 2, n - i - 1]$. From Proposition A.4

$$
\begin{aligned}
\sum_{k=i+2}^{n-i-1} [(x^\star)^\top A]_k &= \sum_{k=i+2}^{n-i-1} x^\star_k = 1 - \left( \sum_{k=1}^{i+1} x^\star_k + \sum_{k=n-i}^{n} x^\star_k \right) \\
&\geq 1 - x^\star_{n-i} - \left( \sum_{k=1}^{i+1} x^\star_j + \sum_{k=n-i+1}^{n} x^\star_j \right) \\
&\geq 1 - (4i + 4)\epsilon - n\epsilon = 1 - (5n + 4)\epsilon
\end{aligned}
$$

Thus, we are ensured that there exists a column $k^\star \in [i + 2, n - i - 1]$ with utility $[(x^\star)^\top A]_{k^\star} \geq \frac{1-(5n+4)\epsilon}{n}$. Now consider the difference between the utility of column $k^\star$ and the column $n - i$.

$$[(x^\star)^\top A]_{k^\star} - [(x^\star)^\top A]_{n-i} \geq \frac{1 - (5n + 4)\epsilon}{n} - 2(4i + 4)^2\epsilon \geq \frac{1 - (5n + 4)\epsilon}{n} - 2(4n + 4)^2\epsilon \geq \epsilon$$

where the last inequality follows by the fact that $\epsilon \leq 1/56n^3$. Hence if the column player puts $y^\star_{n-i}$ probability mass to column $k^\star$ then it increases its payoff by $y^\star_{n-i} \left( [(x^\star)^\top A]_{k^\star} - [(x^\star)^\top A]_{n-i} \right) > \epsilon^2$. The latter contradicts with the assumption that $(x^\star, y^\star)$ is an $\epsilon^2$-NE. Thus we have reached to a final contradiction that $y^\star_{n-i} > \epsilon$.

$\square$

## A.1 Proof of Theorem A.3

**Proposition A.5.** *The sum of utilities of rows* $k \in [i+2, n-i-1]$ *admits,*

$$\sum_{k=i+2}^{n-i-1} [Ay^\star]_k \geq \sum_{k=i+2}^{n-i-1} y_k^\star$$

*Proof.* By Observation 3.3 we derive the following equation.

$$[Ay^\star]_{i+2} = [Ay^\star]_{(i+1)+1} = (4i+2)y_{(i+1)+1}^\star + (4i+3)y_{n-(i+1)}^\star \geq y_{(i+1)+1}^\star$$

The claim can be immediately derived from the inequality given above, $\sum_{k=i+2}^{n-i-1}[Ay^\star]_k \geq \sum_{k=i+2}^{n-i-1} y_k^\star$.

$\square$

## A.2 Proof of Theorem A.4

**Proposition A.6.** *The sum of utilities of columns* $k \in [i+2, n-i-1]$ *admits,*

$$\sum_{k=i+2}^{n-i-1} [(x^\star)^\top A]_k \geq \sum_{k=i+2}^{n-i-1} x_k^\star$$

*Proof.* By Observation 3.3 we derive the following equation.

$$[(x^\star)^\top A]_{i+2} = [(x^\star)^\top A]_{(i+1)+1} \geq (4i+2)x_{(i+1)+1}^\star + (4i+3)x_{n-(i+1)}^\star \geq x_{(i+1)+1}^\star$$

The claim can be immediately derived from the inequality given above, $\sum_{k=i+2}^{n-i-1}[(x^\star)^\top A]_k \geq \sum_{k=i+2}^{n-i-1} x_k^\star$.

$\square$

# B Proof of Lemma 3.8

## B.1 Omitted Proofs of Section 3.3

**Proposition B.1** (Proof of Proposition 3.10)**.** *Let* $(i^{(t)}, j^{(t)})$ *be a strategy selected by fictitious play at round t, and* $(i^{(t)}, j^{(t)}) \neq (\frac{n}{2}, \frac{n}{2})$*. Then, in a subsequent round, fictitious play will choose the strategy of greater value that is either on row* $i^{(t)}$ *or column* $j^{(t)}$*.*

*Proof.* To establish the claim, we employ the concept of a cumulative utility vector, as defined in Definition 2.7. According to Proposition 3.5, the row $i^{(t)}$ and column $j^{(t)}$ combined have three distinct non-zero elements. Without loss of generality, let's assume that the greater element is in column $j^{(t)}$, but in a different row, denoted as $i'$.

Firstly, we observe that any subsequent strategy will involve only those three elements. This is because in the cumulative utility vector, which determines the strategy to be played in each round, only the coordinates corresponding to those elements are updated as long as the strategy $(i^{(t)}, j^{(t)})$ is being played.

Moreover, we can demonstrate that among these elements, the one with the greater value will be played next, and this transition is deterministic. This means that the row player will choose the strategy associated with the greater element. We note that this decision is implicitly affected by the strategy of column player.

Let's first exclude the case where the next strategy switch involves the column player. We initially assumed that the greatest element is in column $j^{(t)}$ but on a different row. Consequently, the only non-zero element in a different column than $j^{(t)}$ must have a smaller utility compared to the element $(i^{(t)}, j^{(t)})$. Therefore, there is no incentive to switch to a strategy with lower utility. This confirms that the column player will not opt for a different strategy, ensuring that the next strategy switch, if it occurs, will necessarily involve the row player.

Regarding the row player, we aim to prove that there will be a round where they will change to a different strategy, and consequently, to a different row. To analyze this, let's examine how the cumulative utility vector of the row player changes from round to round.

$$
i := i^{(t)} \in \underset{i \in [n]}{\operatorname{argmax}} \left[ \sum_{s=1}^{t-1} A e_{j^{(s)}} \right]_i
$$

Therefore, as long as the column player continues to use the same strategy, the row $A e_{j^{(t)}}$ will repeatedly be added to the cumulative vector, reinforcing the coordinate of row $i'$. However, this cannot happen indefinitely, as the cumulative utility vector is bounded. After a certain number of rounds, the row player will eventually choose the strategy associated with row $i'$. This proves that claim for the case of the row player.

Similarly, if the greater element is located in the same row but on a different column, a similar argument can be made to prove that the column player will switch strategies next.

$\square$

**Corollary B.2** (Proof of Corollary 3.11). *Let $t$ be a round in which a player changes their strategy. Then exactly one of the following statements is true:*

1. *If the row player changes their strategy at round t, i.e. $i^{(t)} \neq i^{(t-1)}$, then the column player can only make the next strategy switch.*

2. *If the column player changes their strategy at round t, i.e. $j^{(t)} \neq j^{(t-1)}$, then the row player can only make the next strategy switch.*

*Proof.* This corollary is a simple application of Proposition 3.10. We will only prove the first claim. Let $t$ be the round when the row player changes their strategy, i.e., $i^{(t)} \neq i^{(t-1)}$. According to Proposition 3.5, there are three non-zero elements combined in $i^{(t)}, j^{(t)}$. Since the row player changes their strategy, it follows from Proposition 3.10 that the other element in column $j^{(t)}$ but not in row $i^{(t)}$ must necessarily have a smaller value than $(i^{(t)}, j^{(t)})$. Therefore, the element with the greater value must necessarily be in a different column. Hence, by applying Proposition 3.10, we conclude that the column player can only make the next strategy switch. This proves the claim. $\square$

## B.2 Auxiliary Propositions for Lemma 3.6

In this subsection, we provide the full version of the proposition used to establish the proof Lemma 3.6.

**Proposition B.3.** *There exists a round $T_i^1 > T_i^0$ at which*

(i) *the strategy profile is $(i + 1, i + 1)$ for the first time,*

(ii) *for all rounds $t \in [T_i^0, T_i^1 - 1]$, the strategy profile is $(n - i, i + 1)$,*

(iii) *column $i + 1$ admits cumulative utility $C_{i+1}^{(T_i^1)} \geq (4i + 1) \cdot (R_{i+1}^{(T_i^0)} + 1)$,*

(iv) *all rows $k \in [(i + 1) + 1, n - i - 1]$ admit $R_k^{(T_i^1)} = 0$ and all columns $k \in [i + 2, n - i]$ admit $C_k^{(T_i^1)} = 0$.*

*Proof.* The proposition is composed of multiple parts, each of which is proven separately. To begin with, we must establish that the new strategy profile chosen by fictitious play will be $(i + 1, i + 1)$.

According to the inductive hypothesis in Theorem 3.1, we know that the strategy $(n - i, i + 1)$ was played at round $T_i^0$ for the first time, while any strategy involving a row $\in [i + 1, n - (i + 1)]$ has not been played until that point. Additionally, the inductive hypothesis also states that the strategy played before time $T_i^0$ was $(n - i, n - i + 1)$. Therefore, according to Corollary 3.11, it is guaranteed that the next strategy switch will be initiated by the row player.

In other words, as long as the column player continues to play their current strategy, strategy $(n - i, i + 1)$ will be played in every subsequent round, which establishes Item (ii). This implies that the row player's cumulative utility will increase by $Ae_{i+1}$. As per Observation 3.3, column $i + 1$ only has non-zero elements in positions $i + 1$ and $n - i$.

$$Ae_{i+1} = [0, \ldots, \underbrace{4i + 2}_{i+1}, 0, \ldots, 0, \underbrace{4i + 1}_{n-i}, 0, \ldots] \tag{5}$$

Since strategy $i + 1$ has a higher value, a strategy switch in a later time step is certain. Consequently, there will be a round $T_i^1$ in which the strategy $(i+1, i+1)$ will be played for the first time, establishing Item (i).

We must determine the point at which the strategy switch to $(i + 1, i + 1)$ will take place. According to Equation (5), the value of strategy $i + 1$ is exactly one greater than the value of strategy $n - i$. From the inductive hypothesis, we know that row $i + 1$ has a cumulative utility of zero, whereas row $n - i$ has a cumulative utility of $R_{i+1}^{(T_i^0)}$. Therefore, it will take precisely $R_{i+1}^{(T_i^0)}$ steps for those strategies to have equal cumulative utility values. Consequently, the strategy switch will either occur in that round or the immediate next, as the cumulative utility of row $i + 1$ will have surpassed that of row $n - i$.

In order to proceed with Item (iii), we compute the updated cumulative utility of column $(i + 1)$. As shown in Equation (5), column $i + 1$ has a value of $(4i + 1)$ at position $n - i$. Thus, if row $n - i$ has been played for a minimum of $R_{i+1}^{(T_i^0)}$ rounds, then the cumulative utility of column $i + 1$ is greater than $(4i + 1) \cdot R_{i+1}^{(T_i^0)}$. Given that row $n - i$ has also been played previously (inductive hypothesis), it is reasonable to conclude that:

$$C_{i+1}^{(T_i^1)} \geq (4i + 1) \cdot (R_{i+1}^{(T_i^0)} + 1)$$

Lastly, according to Observation 3.4, there exists a non-zero element in row $i + 1$ at position $i + 1$. Due to the fact that column $i + 1$ has already been played (according to Item (ii)), we can conclude that the cumulative utility of row $i + 1$ must be non-zero. Combining this with inductive hypothesis, it concludes the proof of Item (iv). $\square$

### B.2.1 Proof of Theorem 3.13

**Proposition B.4.** *There exists a round $T_i^2 > T_i^1$ at which*

    *(i) the strategy profile is $(i + 1, n - i)$ for the first time,*

    *(ii) for all rounds $t \in [T_i^1, T_i^2 - 1]$, the strategy profile is $(i + 1, i + 1)$,*

    *(iii) row $i + 1$ admits cumulative utility $R_{i+1}^{(T_i^2)} \geq (4i + 2) \cdot C_{i+1}^{(T_i^1)}$,*

    *(iv) all rows $k \in [(i+1)+1, n-i-1]$ admit $R_k^{(T_i^2)} = 0$ and all columns $k \in [i+2, n-(i+1)]$ admit $C_k^{(T_i^2)} = 0$.*

*Proof.* We repeat the same reasoning as in the proof of Proposition 3.12. To begin with, we must establish that the new strategy profile chosen by fictitious play will be $(i + 1, n - i)$.

We can infer from both Items (i) and (ii) in Proposition 3.12 that the most recent strategy switch was made by the row player. Therefore, based on Corollary 3.11, we are certain that the next strategy switch will be initiated by the column player.

In other words, as long as the row player continues to play their current strategy, strategy $(i+1, i+1)$ will be played in every subsequent round, which establishes Item (ii). This implies that the column player's cumulative utility will increase by $e_{i+1}^\top A$. As per Observation 3.3, row $i+1$ only has non-zero elements at positions $i+1$ and $n-i$.

$$e_{i+1}^\top A = [0, \ldots, \underbrace{4i+2}_{i+1}, 0, \ldots, 0, \underbrace{4i+3}_{n-i}, 0, \ldots] \tag{6}$$

Since strategy $n-i$ has a higher value, a strategy switch in a later time step is certain. Consequently, there will be a round $T_i^2$ in which the strategy $(i+1, n-i)$ will be played for the first time, establishing Item (i).

We must determine the point at which the strategy switch to $(i+1, n-i)$ will take place. According to Equation (6), the value of strategy $n-i$ is exactly one greater than the value of strategy $i+1$. From the Proposition 3.12, we know that column $n-i$ has a cumulative utility of zero, whereas column $i+1$ has a cumulative utility of $C_{i+1}^{(T_i^1)}$. Therefore, it will take precisely $C_{i+1}^{(T_i^1)}$ steps for those strategies to have equal cumulative utility values. Consequently, the strategy switch will either occur in that round or the immediate next, as the cumulative utility of column $n-i$ will have surpassed that of column $i+1$.

In order to proceed with Item (iii), we compute the updated cumulative utility of row $(i+1)$. As shown in Equation (6), row $i+1$ has a value of $(4i+2)$ at position $i+1$. Thus, if column $i+1$ has been played for a minimum of $C_{i+1}^{(T_i^1)}$ rounds, then the cumulative utility of row $i+1$ satisfies $R_{i+1}^{(T_i^2)} \geq (4i+2) \cdot C_{i+1}^{(T_i^1)}$.

Lastly, according to Observation 3.4, there exists a non-zero element in column $n-i$ at position $i+1$. Due to the fact that row $i+1$ has already been played (according to Item (ii)), we can conclude that the cumulative utility of column $n-i$ must be non-zero. Combining this with Proposition 3.12, it concludes the proof of Item (iv). □

### B.2.2  Proof of Theorem 3.14

**Proposition B.5.** *There exists a round $T_i^3 > T_i^2$ at which*

(i) *the strategy profile is $(n-(i+1), n-i)$ for the first time,*

(ii) *for all rounds $t \in [T_i^2, T_i^3 - 1]$, the strategy profile is $(i+1, n-i)$,*

(iii) *column $n-i$ admits cumulative utility $C_{n-i}^{(T_i^3)} \geq (4i+3) \cdot R_{i+1}^{(T_i^2)}$,*

(iv) *all rows $k \in [(i+1)+1, n-(i+1)-1]$ admit $R_k^{(T_i^3)} = 0$ and all columns $k \in [i+2, n-(i+1)]$ admit $C_k^{(T_i^3)} = 0$.*

*Proof.* We repeat the same reasoning as in the proof of Proposition 3.12. To begin with, we must establish that the new strategy profile chosen by fictitious play will be $(n-i-1, n-i)$.

We can infer from both Items (i) and (ii) in Proposition 3.13 that the most recent strategy switch was made by the column player. Therefore, based on Corollary 3.11, we are certain that the next strategy switch will be initiated by the row player.

In other words, as long as the column player continues to play their current strategy, strategy $(i+1, n-i)$ will be played in every subsequent round, which establishes Item (ii). This implies that the row player's cumulative utility will increase by $Ae_{n-i}$. As per Observation 3.3, column $n-i$ only has non-zero elements at positions $i+1$ and $n-(i+1)$.

$$Ae_{n-i} = [0, \ldots, \underbrace{4i+3}_{i+1}, 0, \ldots, 0, \underbrace{4i+4}_{n-(i+1)}, 0, \ldots] \tag{7}$$

Since strategy $n - (i + 1)$ has a higher value, a strategy switch in a later time step is certain. Consequently, there will be a round $T_i^3$ in which the strategy $(n - i - 1, n - i)$ will be played for the first time, establishing Item (i).

We must determine the point at which the strategy switch to $(n - i - 1, n - i)$ will take place. According to Equation (7), the value of strategy $n - (i + 1)$ is exactly one greater than the value of strategy $i + 1$. From the Proposition 3.13, we know that row $n - (i + 1)$ has a cumulative utility of zero, whereas row $i + 1$ has a cumulative utility of $R_{i+1}^{(T_i^2)}$. Therefore, it will take precisely $R_{i+1}^{(T_i^2)}$ steps for those strategies to have equal cumulative utility values. Consequently, the strategy switch will either occur in that round or the immediate next, as the cumulative utility of row $n - (i + 1)$ will have surpassed that of row $i + 1$.

In order to proceed with Item (iii), we compute the updated cumulative utility of column $n - i$. As shown in Equation (7), column $n - i$ has a value of $(4i + 3)$ at position $i + 1$. Thus, if row $i + 1$ has been played for a minimum of $R_{i+1}^{(T_i^2)}$ rounds, then the cumulative utility of column $n - i$ satisfies $C_{n-i}^{(T_i^3)} \geq (4i + 3) \cdot R_{i+1}^{(T_i^2)}$.

Lastly, according to Observation 3.4, there exists a non-zero element in row $n - (i + 1)$ at position $n - i$. Combining this with the Proposition 3.13 and the fact that column $n - i$ has already been played (according to Item (ii)), we can conclude that the cumulative utility of row $n - (i + 1)$ must be non-zero. This concludes the proof of Item (iv).

$\square$

### B.2.3 Proof of Theorem 3.15

**Proposition B.6.** *There exists a round $T_i^4 > T_i^3$ at which*

  (i)  *the strategy profile is $(n - (i + 1), (i + 1) + 1)$ for the first time,*

  (ii)  *for all rounds $t \in [T_i^3, T_i^4 - 1]$, the strategy profile is $(n - (i + 1), n - i)$,*

  (iii)  *row $n - i - 1$ admits cumulative utility $R_{n-(i+1)}^{(T_i^4)} \geq (4i + 4) \cdot C_{n-i}^{(T_i^3)}$,*

  (iv)  *all rows $k \in [(i + 1) + 1, n - (i + 1) - 1]$ admit $R_k^{(T_i^4)} = 0$ and all columns $k \in [(i + 1) + 2, n - (i + 1)]$ admit $C_k^{(T_i^4)} = 0$.*

*Proof.* We repeat the same reasoning as in the proof of Proposition 3.12. To begin with, we must establish that the new strategy profile chosen by fictitious play will be $(n - (i + 1), (i + 1) + 1)$.

We can infer from both Items (i) and (ii) in Proposition 3.14 that the most recent strategy switch was made by the row player. Therefore, based on Corollary 3.11, we are certain that the next strategy switch will be initiated by the column player.

In other words, as long as the row player continues to play their current strategy, strategy $(n - (i + 1), n - i)$ will be played in every subsequent round, which establishes Item (ii). This implies that the column player's cumulative utility will increase by $e_{n-(i+1)}^\top A$. As per Observation 3.3, row $n - (i + 1)$ only has non-zero elements at positions $(i + 1) + 1$ and $n - (i + 1)$.

$$e_{n-(i+1)}^\top A = [0, \ldots, \underbrace{4i + 5}_{(i+1)+1}, 0, \ldots, 0, \underbrace{4i + 4}_{n-(i+1)-1}, 0, \ldots] \tag{8}$$

Since strategy $(i + 1) + 1$ has a higher value, a strategy switch in a later time step is certain. Consequently, there will be a round $T_i^4$ in which the strategy $(n - (i + 1), (i + 1) + 1)$ will be played for the first time, establishing Item (i).

We must determine the point at which the strategy switch $(n - (i + 1), (i + 1) + 1)$ will take place. According to Equation (8), the value of strategy $(i + 1) + 1$ is exactly one greater than the value of strategy $n - i$. From the Proposition 3.14, we know that column $(i + 1) + 1$ has a cumulative utility of zero, whereas column $n - i$ has a cumulative utility of $C_{n-i}^{(T_i^3)}$. Therefore, it will take precisely

$C_{n-i}^{(T_i^3)}$ steps for those strategies to have equal cumulative utility values. Consequently, the strategy switch will either occur in that round or the immediate next, as the cumulative utility of column $(n-(i+1),(i+1)+1)$ will have surpassed that of column $n-i$.

In order to proceed with Item (iii), we compute the updated cumulative utility of row $n-(i+1)$. As shown in Equation (8), row $n-(i+1)$ has a value of $(4i+4)$ at position $n-i$. Thus, if column $n-i$ has been played for a minimum of $C_{n-i}^{(T_i^3)}$ rounds, then the cumulative utility of row $n-(i+1)$ satisfies $R_{n-(i+1)}^{(T_i^4)} \geq (4i+4) \cdot C_{n-i}^{(T_i^3)}$.

Lastly, according to Observation 3.4, there exists a non-zero element in column $(i+1)+1$ at position $n-(i+1)$. Combining this with the Proposition 3.14 and the fact that row $n-(i+1)$ has already been played (according to Item (ii)), we can conclude that the cumulative utility of column $(i+1)+1$ must be non-zero. This concludes the proof of Item (iv). □

