# OpenReview forum: "Exponential Lower Bounds for Fictitious Play in Potential Games"
_NeurIPS.cc/2023/Conference — NeurIPS 2023 poster_

### Official Review · Reviewer_a6So · 2023-06-22

**Soundness:** 4 excellent
**Presentation:** 4 excellent
**Contribution:** 3 good
**Rating:** 7
**Confidence:** 4

**Summary:**

This paper studies fictitious play in potential games and in particular, two-player identical payoff games. It is shown that fictitious play, under arbitrary tie-breaking, needs super-exponential time with respect to the number of actions to find an approximate Nash equilibrium. The lower bound is proved through recursive construction of a hard game in which fictitious play spirals slowly and takes super exponential time to reach the unique pure Nash equilibrium.

**Strengths:**

- This paper makes significant progress on a fundamental problem, namely the convergent behavior of fictitious play in potential games. While Monderer et al. proved an asymptotic $O(1/t)$ rate, no finite-time guarantee is known before to my knowledge; this paper shows that any finite-time guarantee for classes of games that include identical-payoff games has to be super-exponential in the number of actions. It suggests that fictitious play in potential games, while asymptotically convergent, would have a burn-in period that is super-exponentially long.

- The paper is remarkably well written. The results are presented clearly and the construction is clean and easy to follow.

**Weaknesses:**

- The asymptotic part of the lower bound translates to $\Omega(1/(n^2t^2))$, which I believe is not tight in light of the $O(1/t)$ upper bound.


**Questions:**

A couple of minor questions:

- Line 221: In Eq (2), should the last term be $R^{T_{l-1}}_{n-l+1}$ instead?

- Line 247: To evoke Lemma 3.6, wouldn't one need $n\sqrt{\epsilon}<1$? I can't seem to find where this is assumed.

- Line 248: "On the other hand, at each round $t\ge T^*$, ..., with rate $1/t$". Where is result statement proven?

**Limitations:**

This work is of theoretical nature and there is no foreseeable negative ethical, societal implication.

---

> ### Author Rebuttal · Authors · 2023-08-08
>
> We thank the reviewer for all its effort, the insightful comments and the carefully reading of our work. We start by answering the reviewer's questions.
>
> *Questions*
>
> 1. Thanks for spotting the typo.
> 2. The reviewer is right we missed the assumption in Theorem 3.1. In fact Lemma 3.6 requires $\epsilon \in O(1/n^3)$. Since in the proof of Theorem 3.1, Lemma 3.6 is applied for accuracy $\sqrt{\epsilon}$, we need to take $\epsilon < O(1/n^6)$. We will add the latter in the Theorem 3.1.
> 3. Notice that once $(i(t),j(t)) = (n/2,n/2+1)$ for $t = T^\star$ then $(i(t),j(t)) = (n/2,n/2+1)$ for all $t \geq T^\star$ since $(n/2,n/2+1)$ is the maximum entry of the matrix $A$ and thus $n/2$ is the action with most cumulative payoff for agent $x$ (resp. $n/2+1$ for agent $y$). We will add the above explanation in the revised version.
>
> $$ $$
> Concerning the reviewer's major point, we want to remark that our results imply exponential lower bounds even for the $1/t$ case. More precisely, let us assume that an upper bound of the form $p_1(n) + p_2(n)/\epsilon$ could be established. By taking $\epsilon = O(1/n^6)$, Theorem~3.1 applies and thus
> $$ p_1(n) + O(n^6 p_2(n)) \geq 4^n ((n/2-2)!)^4$$
>
> meaning that either $p_1(n)$ or $p_2(n)$ is super-exponential. We will add the above discussion in the revised version of our work.

---

> > ### Comment · Reviewer_a6So · 2023-08-16
> >
> > I thank the authors for answering my questions.
> >
> > **Re: Asymptotic lower bounds**
> >
> > I think something might be missed here. I understand your argument that either $p_1(n)$ or $p_2(n)$ is super-exponential in $n$; what I meant was that an upper bound of the form $p_1(n) + n^2/t^2$ cannot be ruled out by Theorem 1 ($p_1(n)$ being super-exponential in $n$), and that this is not *asymptotically tight* due to existing upper bounds.

---

> > > ### Author Response · Authors · 2023-08-16
> > >
> > > Thank you for the clarification. Providing lower bounds that asymptotically match the $O(1/t)$ rates is a great question for future work.

---

### Official Review · Reviewer_nmF9 · 2023-06-27

**Soundness:** 3 good
**Presentation:** 3 good
**Contribution:** 3 good
**Rating:** 7
**Confidence:** 3

**Summary:**

This paper examines the convergence rate of Fictitious Play (FP) in potential games. The paper proves that FP for potential games can take exponential time to reach a Nash equilibrium. The research has yielded a recursive rule for constructing payoff matrices, demonstrating that fictitious play, regardless of the tie-breaking rule employed, may require exponential time to reach a Nash equilibrium even in two-player identical payoff games. The same theorem holds for general N-player potential games.

**Strengths:**

They have provided a concrete example to illustrate the exponential lower bound with figures, which greatly enhances the clarity and understanding of the paper. Furthermore, I am satisfied with the overall readability of the paper. The writing style is approachable, making it easier for readers to engage with the content. One aspect that I particularly enjoyed is the authors' explanation of intuition to construct counterexamples of learning dynamics. This approach helps readers grasp the underlying concepts more effectively, enhancing the overall learning experience. The proof idea is simple and intuitive.
The paper is well-written, provides clear explanations through intuitive examples, and offers an enjoyable reading experience.

**Weaknesses:**

I have some concerns regarding this paper that I would like the authors to address.

1. In reference [1], it is mentioned that if certain regularity conditions are met, FP dynamics achieve an exponential rate. Can the authors provide more details on these regularity conditions and their implications? Like why this game is not in the regularity conditions. [1] are saying that almost every potential game are regular game.

2. For FP dynamics, this paper proves that finding NE using fictitious play potential games has exponentially lower bounds. Does this exponential lower bound also hold for stochastic fictitious play? For example, if people have some exploration in each turn,  is it possible to break the exponential lower bound?

3. I am curious about the upper-bound of the convergence rate in the case where the conditions of regularity are not satisfied. I guess that the example that this paper gave does not satisfy the regularity condition. Is there a specific upper-bound that can be established?


**Questions:**

I want to clarify the issues of the weakness section. If this is well-addressed, I am down to re-evaluate this paper.



**Limitations:**

The authors adequately addressed the limitation and potential negative societal impact on their work.

---

> ### Author Rebuttal · Authors · 2023-08-08
>
> Thank you for your work and the valuable comments. We would like to address comment 1 and 3 together, as they are closely related. Additionally, we want to clarify that when referring to [1], we assume the paper the Reviewer nmF9 mentioned is titled "On the Exponential Rate of Convergence of Fictitious Play in Potential Games". In comparison to that work, our research presents several important qualitative differences.
>
> We remark that our constructed lower bound *does* satisfy the regularity conditions of [1]. There are two key reasons that [1] and our work are not contradictory. First of all [1] consider the *continuous-time* version of fictitious-play while we consider the *discrete-time* version, thus our lower bound is not necessarily applicable. Secondly and most importantly, [1] establish exponential convergence only in the asymptotic sense. Let us take a closer look to their main theorem.
>
> Theorem 8. Let $\Gamma$ be a regular potential game. Then for almost every initial condition $x_0 \in X$, there exists a constant $c = c(\Gamma, x_0 )$ such that if $x$ is an FP process associated with $\Gamma$ and $x(0)=x_0$ , then $d(x(t), NE) \leq c e^{-t}$.
>
> Notice that $c$ is not a universal constant in fact it is not even polynomially bounded by the size of the game. Meaning that $c$ can have super-exponential dependence on $n$ - for example $c = O(2^{2^{n}})$ or $c = O(4^n((n/2-2)!)^4)$.
>
> In simpler words, [1] establish that for *continuous* fictitious-play there exist a time-step $T^\star$ after which exponential convergence occurs, however $T^\star$ can be super-exponentially large! The latter is totally aligned with our take-way message for the discrete case.
>
> *Comment 2*
>
> Extending our lower bound construction for stochastic fictitious play in an interesting research direction that however remained outside the scope of this work. To this end, we have experimentally studied the convergence properties of stochastic fictitious play in the provided lower bounded construction of this work. Our experimental evaluations are included in the uploaded pdf and suggest that a more fine-grained analysis can be extend our lower bound to stochastic FP.

---

> > ### Comment · Reviewer_nmF9 · 2023-08-10
> > **Very interesting**
> >
> > For both points, I am very satisfied. Thank you for the detailed explanation. Every question that I had is now resolved.
> >
> > I want to adjust my score from 5 to 7.

---

> > > ### Author Response · Authors · 2023-08-10
> > > **Score adjusted?**
> > >
> > > We thank the reviewer for their answer. May we ask if the reviewer adjusted the score because it still appears to be a 5. Thank you again.

---

> ### Comment · Reviewer_nmF9 · 2023-08-13
> **Just came in my mind**
>
> I adjusted the score to 7.
>
> But while thinking about this topic, a question came to my mind: so are there no polynomial algorithms for finding NE in potential games? I think I saw several papers that have policy gradient (or gradient descent), but have some assumptions on the exploration. Or, if we have an assumption about exploration (like many papers assume for the zero-sum game), can we achieve polynomial complexity for the FP algorithm?

---

> > ### Comment · Reviewer_nmF9 · 2023-08-13
> > **Other question**
> >
> > I am writing this because the authors are experts on the fictitious play dynamics domain.
> > So, one more additional question:  is there an alternative convergence rate that has a polynomial dependency on $n$ while having an exponential dependency on $t$?
> >
> > (Disclaimer: these two questions will not make a lower score even if the authors do not have any explicit and clear answers. Just I am curious about this.)

---

> > > ### Author Response · Authors · 2023-08-14
> > > **Replying to "Other question"**
> > >
> > > To the best of our knowledge no such convergence rate for FP exist.

---

> > ### Author Response · Authors · 2023-08-14
> > **Relying to "Just came in my mind"**
> >
> > Thank you very much for your response and for updating your score.
> >
> > *Concerning your question*
> >
> > Finding NE for potential games with a *centralized* algorithm is a computationally easy task (consider the pure strategy profile corresponding to the maximum entry on the game). However establishing convergence to NE with *uncoupled learning dynamics* is way more challenging setting. To the best of our knowledge the polynomial convergence rates exist only for GD-based dynamics e.g. [1]. There are also works establishing asymptotic convergence for MWU in potential games e.g. [2]. We believe that even under exploration assumption, FP cannot achieve polynomial complexity for potential games. The latter is also indicated by the experimental evaluations that we provided in our initial response.
> >
> > [1] Global Convergence of Multi-Agent Policy Gradient in Markov Potential Games, Leonardos et al. 2022
> >
> > [2] Learning with Bandit Feedback in Potential Games, Heliou et al. 2017

---

### Official Review · Reviewer_hanH · 2023-06-30

**Soundness:** 4 excellent
**Presentation:** 3 good
**Contribution:** 3 good
**Rating:** 7
**Confidence:** 4

**Summary:**

This paper constructs a common-payoff two-player game for which fictitious play must take a number of iterations exponential in the number of actions to converge to an $\epsilon$-equilibrium.  The paper closes with some empirical demonstrations of the behavior of fictitious play on the game.

**Strengths:**

The construction of $A$ is very clever, and the description of why this guarantees an exponential lower bound does an excellent job of delivering the important aspects without getting bogged down in the arithmetic details.

Fictitious play is an extremely well-studied and widely-used algorithm.  The structural simplicity of the constructed example gives an excellent intuition about a class of scenarios in which we should expect FP to perform poorly.

**Weaknesses:**

My only concern about this work is that it might have limited impact; a link to the structural properties of games that FP is commonly applied to might have helped.

**Questions:**

1. What is $z$ for in the construction of the game?  Once $K_n(z)$ has been defined, we only ever think about $A=K_n(0)$ for the rest of the paper.


#### Minor issues (did not affect my evaluation)
- on p.3, $x^\top A(y') - x^\top Ay = \Phi(x,y') - \Phi(x,y)$ should be $x^\top B(y') - x^\top By = \Phi(x,y') - \Phi(x,y)$
- on p.7, "Applying Lemma 3.6 for $\epsilon := \sqrt\epsilon$...": This is needlessly confusing

---

> ### Author Rebuttal · Authors · 2023-08-08
>
> Thank you for your work and your feedback on our paper. We appreciate your valuable comments.
>
> 1. Indeed, A should have been B; thank you for pointing it out.
>
> 2. The equality sign does indeed get confusing for the reader; we will change it to either $\coloneqq$ or $\rightarrow$ to signify assignment rather than equality.
>
> 3. $z$ is just a positive real number defining the matrix $K_n(z)$, see Figure 1a. In Definition~3.2 $K^n(z)$ is recursively defined and the construction of our game corresponds to $A = K_n(0)$.
>
> Regarding the potential limited impact of our work: Our study contributes to the existing literature by investigating the convergence rate of Fictitious Play in potential games. Fictitious Play (FP) is a fundamental concept in understanding learning dynamics within strategic interactions, originally introduced by Brown in 1951 [1]. While previous work by Robinson [2] established convergence in zero-sum games, it didn't provide a precise rate of convergence (Karlin conjectured that the rate of FP in zero-sum games is of order $O(1/\sqrt{T})$). FP possesses not only an intuitive game theoretic interpretation but also other attractive properties, such as its simplicity with no need for a step-size parameter. These traits have contributed to FP's status as a pivotal topic of study, leading to numerous works aiming to establish convergence in broader settings [8, 9, 10, 11].
>
> Furthermore, it's essential to emphasize that FP, in conjunction with Blackwell's approachability theorem [6], laid the groundwork for the development of no-regret learning algorithms [15, 21], a thriving area of research. Additionally, the stochastic version introduced by Hofbauer and Sandholm [12] inspired the well-established Follow the Perturbed Leader (FTPL) dynamics [13].
>
> The relevance of theoretical investigations into the convergence of FP has been amplified in recent years. Notably, Daskalakis et al. [3] and Abernethy et al. [4] addressed Karlin's conjecture, a long-standing problem dating back to 1959 [5]. Similarly, Monderer et al.'s foundational work [7] demonstrated FP's convergence in potential games, yet without providing a specific rate of convergence. It was this particular gap in knowledge that sparked our investigation. This work builds upon these preceding studies and addresses a specific aspect that remained unexplored.
>
> There are also many works in the literature published in different venues such as ML/AI [20, 22, 23, 24], theoretical computer science [1, 2, 3, 4, 10, 11, 21], and control theory  [16, 17, 18, 19]. DeepMind used an algorithm called Prioritized Fictitious Self Play as part of the training for their AlphaStar program for playing competitive Starcraft [14].
>
> In light of this, our research contributes to this lineage of ideas, providing insights into the convergence behavior of Fictitious Play in potential games and augmenting the broader landscape of game theory research. We believe our work has meaningful implications for understanding learning dynamics and strategic interactions, and we hope the reviewer finds our perspective on its potential impact clearer in light of these connections.
>
> [1] Iterative solution of games by fictitious play, Brown et al.
>
> [2] An iterative method of solving a games, Robinson et al.
>
> [3]  A Counter-Example to Karlin's Strong Conjecture for Fictitious Play, Daskalakis et al.
>
> [4] Fast Convergence of Fictitious Play for Diagonal Payoff Matrices, Abernethy et al.
>
> [5] Mathematical Methods and Theory in Games, Programming, and Economics, Karlin
>
> [6] An analog of the minimax theorem for vector payoffs, Blackwell
>
> [7] Fictitious Play Property for Games with Identical Interests, Monderer et al.
>
> [8] On the convergence of the learning process in a 2 x 2 non-zero-sum two-person game, Miyasawa et al.
>
> [9] Some topics in two-person games, Shapley et al.
>
> [10] A 2 × 2 game without the fictitious play property, Monderer et al.
>
> [11] On the rate of convergence of fictitious play, Brandt et al.
>
> [12] On the Global Convergence of Stochastic Fictitious Play, Hofbauer et al.
>
> [13] Efficient algorithms for online decision problems, Kalai et al.
>
> [14] Grandmaster level in StarCraft II using multi-agent reinforcement learning, Vinyals et al.
>
> [15] Prediction,learning, and games, Cesa-Bianchi et al.
>
> [16] Joint strategy fictitious play with inertia for potential games, Marden et al.
>
> [17] Forecasting interactive dynamics of pedestrians with fictitious play,  Wei-Chiu et al.
>
> [18] Fictitious play in zero-sum stochastic games, Sayin et al.
>
> [19] Approximation guarantees for fictitious play, Conitzer et al.
>
> [20] Smooth Fictitious Play in Stochastic Games with Perturbed Payoffs and Unknown Transitions, Baudin et al.
>
> [21] No-regret dynamics and fictitious play, Viossat et al.
>
> [22] Fictitious play for mean field games: Continuous time analysis and applications, Perrin et al.
>
> [23] Fictitious play and best-response dynamics in identical interest and zero-sum stochastic games, Baudin et al.
>
> [24] Provably efficient fictitious play policy optimization for zero-sum Markov games with structured transitions,  Qiu et al.

---

> > ### Comment · Reviewer_hanH · 2023-08-14
> >
> > Thanks for your response!  I'm convinced that this work is an important contribution to the literature, and I will update my rating accordingly.

---

> > > ### Author Response · Authors · 2023-08-19
> > >
> > > Thank you very much for your response and for increasing your score.

---

### Official Review · Reviewer_TVCP · 2023-07-08

**Soundness:** 3 good
**Presentation:** 4 excellent
**Contribution:** 3 good
**Rating:** 7
**Confidence:** 3

**Summary:**

This paper studies the convergence rate of Fictitious Play (FP) in potential games. While prior work shows that FP asymptotically converges to a NE in the case of $N$-player potential games, the current paper proves that FP can take exponential time to do so. Specifically, the paper recursively constructs a two-player coordination game with identical interests and a unique pure NE. The paper shows that FP requires super-exponential time before placing positive probability in the unique NE, even with an arbitrary tie-breaking rule. The paper also contains simulations to validate the shown learning dynamics.

**Strengths:**

1. The paper provides a complete answer to an important problem (convergence rate of FP in potential games).
2. The paper is very well-written and easy to follow. Related works are discussed extensively. Technical results are delivered in a clear way.
3. The paper also provides simulations to validate findings and provide intuition.

**Weaknesses:**

1. It might be better to include more background information on FP and potential games, e.g., some concrete examples of potential games in practice.

**Questions:**

Line 58 - 60: I believe [1] is only for diagonal games?

Line 100 - 101: The latter A should be B.

**Limitations:**

N/A.

---

> ### Author Rebuttal · Authors · 2023-08-08
>
> We would like to thank the reviewer of all the work. We sincerely appreciate the reviewer's valuable comments. Related to reviewer's questions/concerns:
>
> 1. $[1]$ is for diagonal zero-sum games and moreover the tie-breaking rule is fixed in advance (not adversarial as in [2]).
>
> 2. Indeed, $A$ should have been $B$; thank you for pointing it out.
>
> 3. We will incorporate an example of a potential game in the preliminaries section and expand our references. Potential games are notably important in capturing routing games (also known as congestion games [3,4]).
>
> [1] Fast Convergence of Fictitious Play for Diagonal Payoff Matrices, Abernethy et al.
>
> [2] A Counter-Example to Karlin's Strong Conjecture for Fictitious Play, Daskalakis et al.
>
> [3] A class of games possessing pure-strategy Nash equilibria, Rosenthal.
>
> [4] Potential games, Monderer et al.

---

> > ### Comment · Reviewer_TVCP · 2023-08-15
> >
> > I thank the authors for the response. I will keep my positive score and vote for acceptance.

---

> > > ### Author Response · Authors · 2023-08-19
> > >
> > > Thank you again for all your work and the positive feedback on our paper.

---

### Author Rebuttal · Authors · 2023-08-08

We thank the reviewers for their hard work. We have attached a pdf with experimental evaluations on the stochastic version of fictitious play, to address a question asked by reviewer nmF9 (what happens in our constructed game if each agent has some exploration). The experiments suggest that a more fine-grained analysis can extend our lower bound to stochastic FP.

---

### Decision · Program_Chairs · 2023-09-21

**Decision:**

Accept (poster)

**Comment:**

After some discussion, all reviewers agreed that the paper should be accepted. The authors should incorporate their replies to the reviewers about related work into the main body of the paper, which will help clarify the significance of their theorem.